



# Folding due to anisotropy in ice, from drill core-scale cloudy bands to km-scale internal reflection horizons

Paul D. Bons[1,2], Yuanbang Hu[1,3], M.-Gema Llorens[4], Steven Franke[1], Nicolas Stoll[5], Ilka Weikusat,[1,5], Julien Westhoff[6], Yu Zhang[1]

[1]Department of Geosciences, Tübingen University, Tübingen, Germany
[2]China University of Geosciences (Beijing), Beijing, China
[3]Center for High Pressure Science and Technology Advanced Research, Beijing, China
[4]Geosciences Barcelona GEO3BCN-CSIC, Spain
[5]Alfred Wegener Institute, Helmholtz Centre for Polar and Marine Research, Bremerhaven, Germany
[6]Niels Bohr Institute, University of Copenhagen, Copenhagen, Denmark

*Correspondence to*: Paul D. Bons (paul.bons@uni-tuebingen.de)

**Abstract.** Upright folds in ice sheets are observed on the cm-scale in cloudy bands in drill cores and on the km-scale in radargrams. We address the question of the folding mechanism for these folds, by analysing the power spectra of fold trains to obtain the amplitude as a function of wavelength signal. Classical Biot-type buckle folds due to a rheological contrast between layers develop a characteristic wavelength, visible as a peak in the power spectrum. Power spectra of ice folds, however, follow a power law with a steady increase in amplitude with wavelength. Such a power spectrum is also observed in a folded, highly anisotropic biotite schist and in a numerical simulation of the deformation of ice Ih with a strong alignment of the basal planes parallel to the shortening direction. This suggests that the folds observed in ice are primarily due to the strong mechanical anisotropy of ice that tends to have a strong lattice preferred orientation due to ice-sheet flow.

## 1 Introduction

Folds are observed on all scales in glaciers and ice sheets. Large-scale folds (100-1000 m scale) are observed via internal reflection horizons (IRHs) in radargrams (Wolovick et al., 2014; Bell et al., 2014; Panton and Karlsson, 2015; Leysinger-Vieli et al., 2018; NEEM community members, 2013; Bons et al., 2016; Franke et al., 2023; Jansen et al., 2024) and in satellite images of the ice surface in West Greenland. Folds on the intermediate scale (~m scale) are common in glaciers (Hudleston, 2015), but more difficult to observe in ice sheets because of the snow cover. Small-scale folds (≤1 cm) in ice cores are visible as undulated cloudy bands, thin layers of elevated impurity concentration mainly occurring in glacial periods (Thorsteinsson, 1996; Alley et al., 1997; Svensson et al., 2005; Faria et al., 2010; Fitzpatrick et al., 2014; Jansen et al., 2016; Weikusat et al. 2017; Stoll et al., 2023). These folds are the main topic of this paper, using examples from the EGRIP drill core (Westhoff et al., 2021 (Fig. 1), that provided novel insights into the crystal orientation inside an ice stream (Stoll et al., 2024), i.e. the Northeast Greenland Ice Stream (NEGIS). "Cloudy bands" made visible in dark-field macroscopy in EGRIP ice are observed already in ice originating from the Younger Dryas at 1257 m of depth (Bohleber et al., 2022),but are a recurring stratigraphic



feature from a depth of 1375 m (Westhoff et al., 2021, Stoll et al., 2023). Chemical data from these bands show elevated impurity concentration and more insoluble particles than in the surrounding layers (Bohleber et al., 2022, Stoll et al., 2023). Stoll et al. (2023) define different cloudy band types and discuss their formation, but little is known about the folding mechanism of these bands that are observed below 1375 m at EGRIP. Folds are not always observed below this depth, which can be explained by the orientation of the drill-core section relative to the fold axis. Only sections at a large angle to the fold axis will reveal folds in the cloudy bands (Figure 4 in Westhoff et al., 2021).


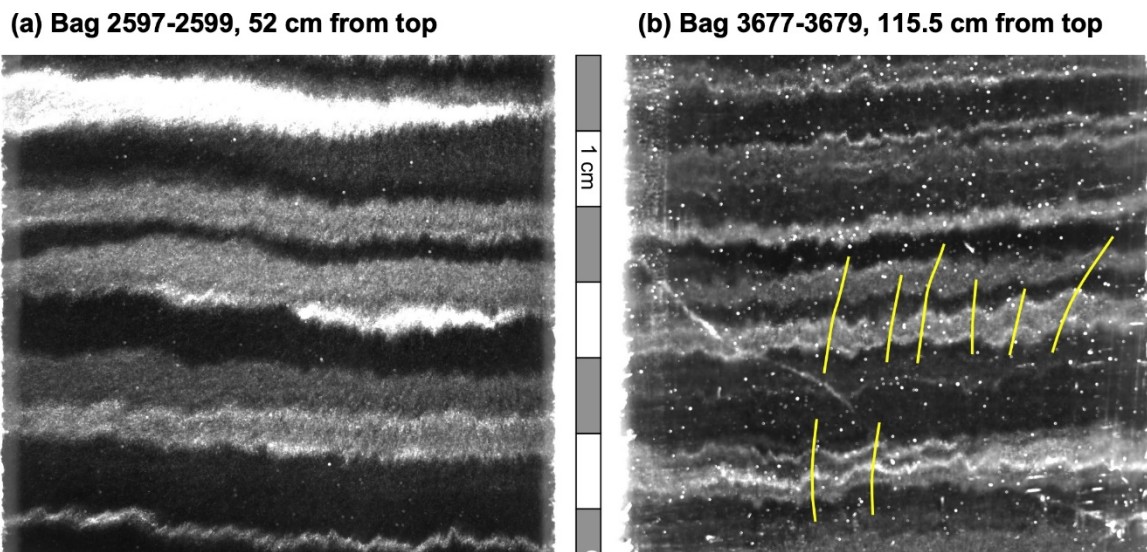

**Figure 1. Two visual stratigraphy line scan examples of folded cloudy bands in the EGRIP drill core from a depth of (a) 1427.8 m and (b) 2017.45 m. Cloudy bands vary in thickness from about one to more than 10 mm. Eight axial planes are drawn as yellow lines in (b). These show that the folds are upright or moderately inclined. The folds are**
**disharmonic, as axial planes cannot be traced for over more than a few times the fold wavelength at the most.**

Nabavi and Fossen (2021) define folds as "curviplanar structures that form by transformation of any tectonic or primary foliation into curved geometries through a non-linear transformation". In geology, 'foliation' is used to denote any pervasive planar structure in a rock (which includes ice). The primary foliation in glaciers and ice sheets is the original layering formed 50 by the deposition of snow layers on the surface. Other foliations can, for example, be healed fractures or fractures filled with frozen water (Hudleston, 2015). Numerous causes for folding in ice sheets have been proposed. Variations in bedrock elevation (Krabbendam, 2016), variable bedrock sliding (Wolovick et al., 2014), basal melting or freeze-on (Leysinger-Vieli et al., 2018) have been proposed to explain large-scale folds of the original stratigraphy. Such external causes for folding cannot apply to small to medium-scale folds that are observed throughout the column of the ice sheet well above the ice-bedrock interface.



These folds must, therefore, be the result of an internal response to layer-parallel shortening (NEEM community members, 2013; Hudleston, 2015; Bons et al., 2016; Jansen et al. 2016; Zhang et al., 2024).

A volume of a mechanically homogeneous and isotropic material will not produce folds when subject to deformation, as the material would only thicken without experiencing localized strain. Folds can form when the material has a mechanical layering

that forms a 'composite anisotropy' and/or when it is 'intrinsically anisotropic', for example due to a crystallographic preferred orientation (CPO) (Griera et al., 2013; Nabavi and Fossen, 2021; Hansen et al., 2021). The latter is often the case in ice because ice normally deforms by dislocation creep (Glen, 1955; Weertman, 1983) that results in a CPO that aligns the easy-glide basal planes in certain preferred orientations (Duval et al., 1983; Budd and Jacka, 1989; Faria et al. 2014; Llorens et al., 2017). In both cases the application of a differential stress will normally lead to a heterogenous deformation field, which implies that

originally straight planar surfaces get distorted: folds develop. Here we will show, based on fold theory and numerical simulations, that folds observed in cloudy bands in the EGRIP drill core primarily result from an intrinsic anisotropy due to the CPO and not from rheological differences between the individual cloudy bands.

## 2 Basic fold terminology and theory

For detailed reviews of fold geometry and terminology the reader is referred to the textbooks of Ramsay and Huber (1987)

and Twiss and Moores (2007), or to the extensive review by Nabavi and Fossen (2021) that also provides an overview of fold theory. Here, we only provide a summary of the relevant terminology and theory based on the above publications, unless otherwise referenced.

Most fold trains roughly resemble a sinusoidal wavefunction (Fig. 2a). One individual fold consists of two limbs that meet at

the fold hinge, the line of maximum curvature. When the fold hinges diverge downwards, the fold is termed an 'antiform', otherwise it is a 'synform'. Antiforms and synforms join at the inflection points in the fold limbs where the direction of curvature changes sign. The terms 'anticline' and 'syncline' are reserved for folds in a stratigraphic sequence and, therefore, apply to folds observed in radargrams or cloudy bands, as these are assumed to represent sedimentary snow layers. Folds can have shapes that range from rectangular boxes, semi-ellipses, parabolas, sine waves, through to chevron or kink folds (Nabavi and Fossen,

2021). Box folds have two-fold hinges per fold, but the authors are unaware of any box folds reported in ice. Ideal chevron or kink folds have straight limbs and highly concentrated curvature in the hinges. Such folds were described in a drilled ice core by Jansen et al. (2016). As most folds resemble a sine wave, the term 'wavelength' ($l$) is one metric used to describe the length scale of folds. It is defined as double the distance between inflection points in the directions of the fold train (Fig. 2a). Accordingly, the 'amplitude' ($A$) is defined as half the distance between the average antiformal and synformal hinge lines,

measured in the direction perpendicular to the fold train. Folds can, however, have multiple wavelengths (Fig. 2b), in which case defining the amplitude becomes difficult. The 'arc length' is the length of a line along the fold trace. The relative arc length



ratio is the ratio of the arc length and the length of a straight line along the fold trace. The arc length is the same as the initial length if a layer only folds and does not become thicker or thinner during folding.


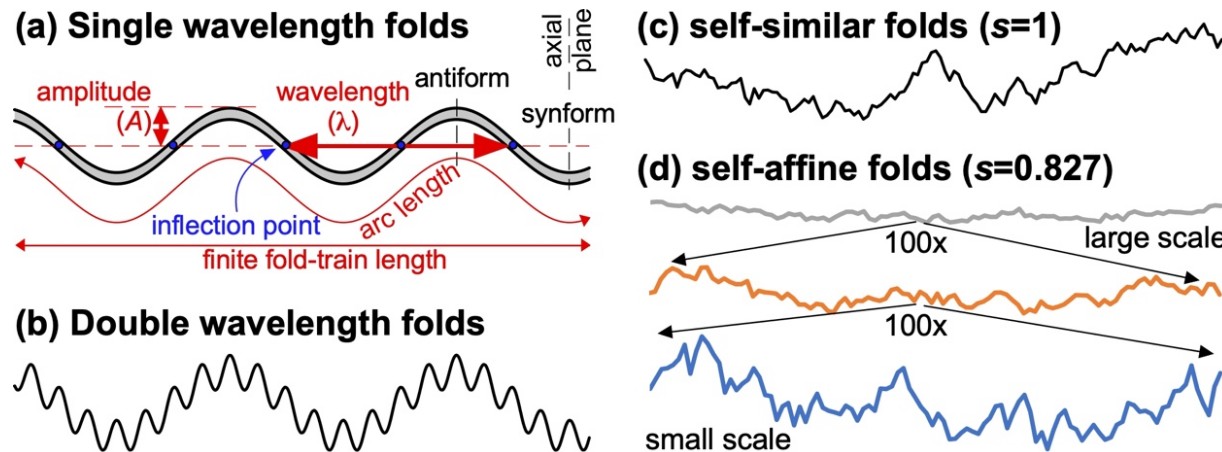

**Figure 2. Fold shape and terminology. (a) Basic fold with a single wavelength. The arc length is the length of the fold train measured along the fold, while the finite length is the distance between the two ends of the fold train. (b) Fold composed of the superposition of two sine waves with different wavelengths and amplitudes. (c) Self similar fold train**

**in which the amplitude of folds is a linear function of the wavelength (*s*=1 in Eq. 2). The artificial fold train is sampled every 0.5 mm if the whole fold train is 65 mm, comparable to the length of fold trains analysed in the EGRIP drill core. (d) Example of a self-affine folds where the amplitude/wavelength ratio systematically decreases with increasing scale (the exponent *s* is defined in Eq. 2).**


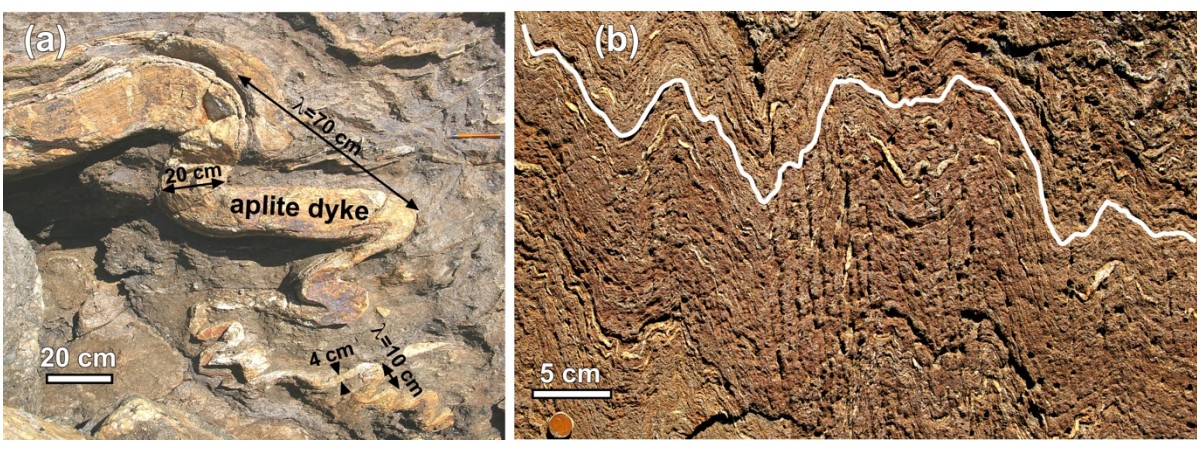



**Figure 3. Examples of folds in rocks from Cap de Creus, northeast Catalonia, Spain. (a) Biot-type folding of a strong aplite dyke in a weaker granodiorite matrix. The example clearly shows the positive correlation between thickness of the dyke and wavelength that both decrease from the top left to the bottom right (Punta Fallarons; N42°20'25", E3°,15'48"). (b) Folded highly anisotropic biotite schist with thin quartz veins showing harmonic multi-wavelength folds. The hand-drawn white line was used for the analysis shown in Fig. 7b (Puig Culip; N42°19'19", E3°18'12").**

Folds may form when a composite material consisting of layers with different rheology is shortened parallel to the layers. The reason for this 'buckle folding' is that it is energetically more favourable to accommodate part of the shortening by bending the stronger layers at intervals (the fold hinges) and rotating the sections in between (the fold limbs). The weaker layers in between need to accommodate this deformation by deforming at a higher rate. Biot (1957) first developed the theory that the final fold wavelength is a function of the amplification rate of an infinite range of wavelengths of initial perturbations in the original layer. The basic idea is that the final wavelength is the one with the highest amplification rate. For a single layer with thickness $H$ and linear viscosity $\eta_l$ embedded in an infinite matrix with viscosity $\eta_m$ he derived for the dominant wavelength $l_d$:

$$\lambda_d = 2\pi H \left(\frac{\eta_l}{6\eta_m}\right)^{1/3} \qquad\qquad (1)$$

Since Biot's pioneering work, many authors extended and refined his theory for multilayers, elasticity, slip or no slip between layers, and for non-linear (power-law) rheologies (see Table 2 in Schmalholz and Mancktelow, 2016). All these theories have in common that when the rheological difference between layers approaches zero, the dominant wavelength reduces to approximately the layer thickness. Wavelengths smaller than the layer thickness cannot be explained by Biot-type buckle-fold theory for layers with different rheological properties. It should also be noted that the fold amplification rate decreases with decreasing rheological contrasts (Llorens, 2019) in isotropic materials. This means that while relatively short wavelengths are possible at low rheological contrast between layers, one would not see the folds as their amplitude would be too small.

Relatively little work has been done on folding due to an intrinsic anisotropy, for example the alignment of easy-glide basal planes in ice or aligned micas in a schist (Cobbold et al., 1971). Biot-type buckle-fold theory cannot be simply applied to predict a dominant wavelength, as there is no layer thickness to provide a length scale. Without a length scale in the system, folds of all wavelengths should amplify at the same rate. Instead of folds with a dominant wavelength, one would expect folds where the amplitude of each wavelength is proportional to that wavelength (Fig. 2c). In that case we get:

$$A_{(\lambda)} = \lambda_0 \cdot \lambda^s, \qquad\qquad (2)$$

with $l_0$ a proportionality constant and $s$ the scaling exponent. When $s=1$ the folds are self-similar, meaning that folds at all scales look similar because the scaling of amplitude and wavelength is identical (Fig. 2c). When $s<1$, large folds have relatively smaller amplitudes than small folds (Fig. 2d). When the scaling of amplitudes and wavelengths is not identical, the folds are self-affine.



## 3 Materials and methods

### 3.1 Materials

Metamorphic schists are mechanically highly anisotropic due to the alignment of platy mica grains into a foliation. Micas are comparable to ice Ih as they deform most easily along their basal planes (Duval et al., 1983; Finch et al., 2021). Metamorphic

schists at Cap de Creus, NE Catalonia, Spain, show folding of the foliation that happened during the Palaeozoic Variscan Orogeny (Druguet et al., 1997, Bons et al., 2004). The mechanical anisotropy of the foliated rock is thought to play a dominant role in the deformation of these rocks (Carreras et al., 2013; de Riese et al., 2019). We therefore use one outcrop as an example of folding of a strongly anisotropic rock in which layering is absent (Fig. 3b).

The East Greenland Ice-core Project (EGRIP) is a deep drilling project located in the middle of NEGIS at 75°37.820 N and 35°59.556 W. Visual stratigraphy line scans of the drill core reveal "cloudy bands" by imaging a polished slab of the core in dark-field macroscopy4. When the section of the core is suitably oriented relative to the flow direction, these cloudy bands show folds (Westhoff et al., 2021) (Fig. 2). We used (Fig. 4) line-scan images 2597_1_32mm.bmp, 3128_1_32mm.bmp, and 23677_1_32mm.bmp (Weikusat et al., 2020), each representing three 55 cm-bags to make a core-length of 165 cm in each

image. The depth of the top of the three images is 1427.8, 1719.85, and 2021.8 m, respectively. We refer to these images by the number of their top bag and depths of individual cloudy bands are given in cm from the top of the image.
Line scan images of bags 2597, 3128 and 3677, ranging in depth from 1428 down to 2022m were used to analyse the folded cloudy bands (Fig. 4)

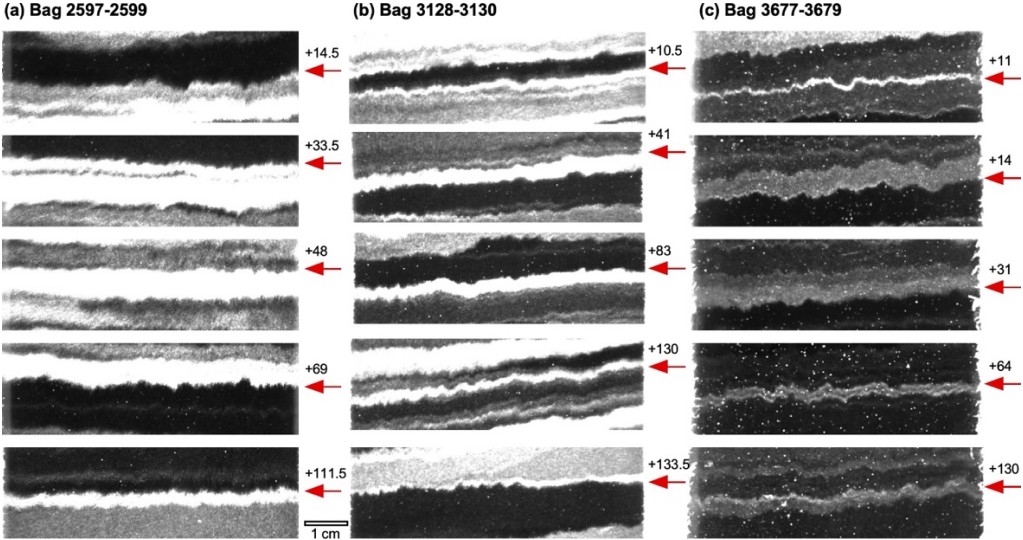


**Figure 4. Images of ten of the 15 cloudy-band interfaces (red arrows) that were analysed in this study. Red arrows indicate the analysed cloudy band interface with distance from top of the line-scan image given in cm.**





160 We use high-resolution radar data from the EGRIP-NOR-2018 survey (Franke et al., 2022b) from the onset region of NEGIS. The radar data were acquired in May 2018 with the AWI's (Alfred Wegener Institute Helmholtz Centre for Polar and Marine Research) airborne multichannel ultra-wideband (UWB) radar and have a horizontal resolution of ~15 m and vertical resolution of 4.31 m. The radargrams used here are centred at the EGRIP drill site and run perpendicular to ice flow (Figure 5).

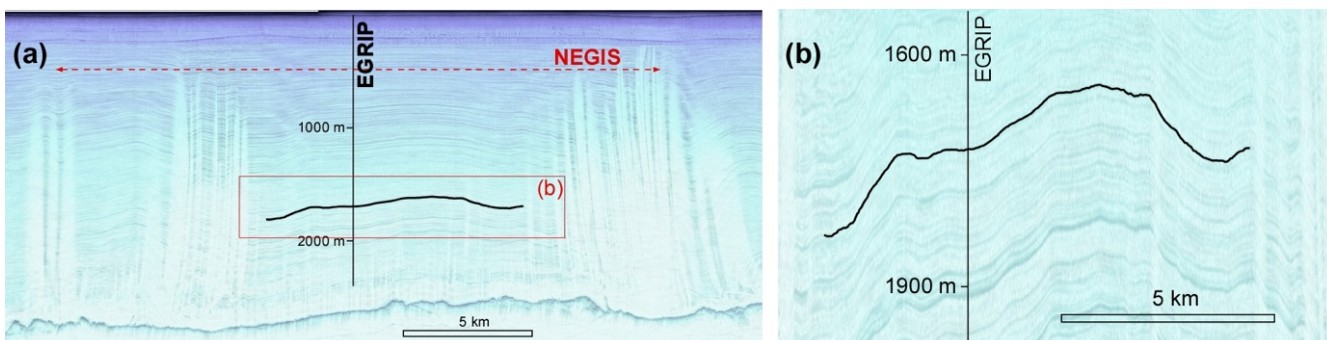

**Figure 5. Traced internal reflection horizon (IRH) in a radargram going through NEGIS perpendicular to ice flow (ice flow into the page) at the location where the EGRIP ice core is drilled. The radargram shown in (a) with a 4.3x vertical exaggeration is composed of profiles 20180508_06_004 and 20180514_03_001 (Franke et al., 2021, 2022b). (b) The same profile showing only the centre of NEGIS at an 18.2x vertical exaggeration at which the layer intersecting EGRIP at 1720 m depth was manually traced.**

## 3.2 Methods

### 3.2.1 Numerical modelling

We use the full-field Viscoplastic Fast Fourier Transform (VPFFT) crystal plasticity code (Lebensohn, 2001; Lebensohn et al., 2008; Lebensohn and Rollett, 2020), coupled with the modelling platform Elle (Bons et al., 2008; Piazolo et al., 2019) to illustrate the fold geometries that form when ice Ih with a mechanically anisotropy is shortened. The code has been used to simulate microstructural developments in deforming ice (Llorens et al., 2016a, 2017; Steinbach et al., 2016) and the formation of folds and other structures in intrinsically anisotropic rocks (Ran et al., 2019; de Riese et al., 2019; Hu et al., 2024). The VPFFT-code simulates deformation of a crystalline material by glide along crystallographic planes. We use the crystallography of hexagonal ice Ih, which is mechanically highly anisotropic due to much easier glide along its basal planes, compared to glide along the prismatic and pyramidal slip systems (Duval et al. 1983). This crystal symmetry approximates a transversely



isotropic material (Griera et al., 2013). We use a stress exponent of four (Goldsby and Kohlstedt, 2001; Bons et al., 2018) for the power-law relation between strain rate and stress and assigned a 16-times higher slip resistance to the non-basal slip systems. At a given strain rate, the stress difference between the basal and non-basal slip systems is thus a factor 16. Details of this modelling approach can be found in Griera et al. (2013) and Llorens et al. (2017).

The 2D models consists of an initially square 256x256 grid of so-called unodes (Bons et al., 2008) that store the local lattice orientation. The unodes effectively represent crystallites or single grains with a constant internal crystal orientation, defined by three Euler angles. Using a Potts model, we created 1995 clusters of identical orientation or grains. On average, each grain is almost 6x6 unodes in size. The basal planes of the initial model were aligned, so that the c-axes normal to the basal planes form a point maximum (with a standard deviation of 10°) parallel to the vertical extension direction. Using velocity boundary conditions, the square model was deformed by horizontal shortening of 2% per calculation step up to 40% shortening, accommodated by vertical stretching.

### 3.2.2 Fold analysis

Between the two end-member fold shapes of box and chevron folds, folds resemble a wave or the addition of multiple waves. Not surprising, Fourier analyses have been applied to folds for about 50 years (Hudleston, 1973; Ramsay and Huber, 1987; Schmalholz and Mancktelow, 2016). This can be used to determine whether the fold train has a single dominant wavelength or is composed of folds of different wavelengths (Fig. 2). We therefore applied a Fast Fourier transform to fold contours of both natural and numerical folds.

We used 2044x31550 8-bit images of the line scans with a resolution of 18.6 pixel/mm from three images. Image 'bag 2597' (Fig. 4a) and 'bag 3128' (Fig. 4b) had a suitable contrast, but a ca. 2x contrast stretch was applied to 'bag 3677' (Fig. 4c) to achieve a sufficient contrast between dark and bright cloudy bands. In each bag five boundaries between dark and bright cloudy bands were selected that were both sharp and where the adjacent cloudy bands showed no significant lateral variation in thickness. A selection of the image was then subjected to a median filter with a 4-pixel radius to reduce small-scale noise and then thresholded to a binary image. The folded trace was subsequently selected by edge detection between the now black and white bands, resulting in the lines shown in Fig. 6a. Only the middle 65 mm of the ca. 70 mm wide drill core image was used to avoid artifacts at the edges of the image. The selection was scaled to 1024 pixels, or 65 mm, width. All this was done with the freeware ImageJ (Schneider et al., 2012). A script selected the $y$-coordinates of the line for each $x$-coordinate along the trace. The equidistant $x,y$-data were then detrended by subtracting a linear least-squares best fit through the $x,y$-data. The detrended series of 1024 $y$-data was then subjected to a discrete Fourier transform using the routine four1() of Press et al. (1992). The power spectrum was obtained by taking the square root of the sum of the squares of the real and imaginary parts of the transform for each wavelength.


For the power spectrum of the numerical folds of Llorens et al. (2013), we applied the above method to a black-and-white image of one of the modelled folds (Fig 6c) in that paper to convert to upper boundary of the folded layer in a set of 1024 equidistant *x,y*-coordinates. For the large-scale folds in NEGIS, we used a radargram that spans NEGIS (Franke et al., 2022b) and is located closely (a few meters) to the EGRIP ice core (Fig. 5) and chose a conspicuous layer at the depth of 'bag 3128' (1719.85 m). The layer was traced for 10 km, only within NEGIS to avoid effects of the higher strain in the shear margins (Jansen et al., 2024). The image on which the layer was traced had a 18.2x vertical exaggeration. For the folded schist we used a 3008x2000 pixel field photograph (Fig. 3b). In each case the selected folded surface was hand-traced in a drawing program (Canvas12) to create a line that was further processed the same way as the cloudy band interfaces.

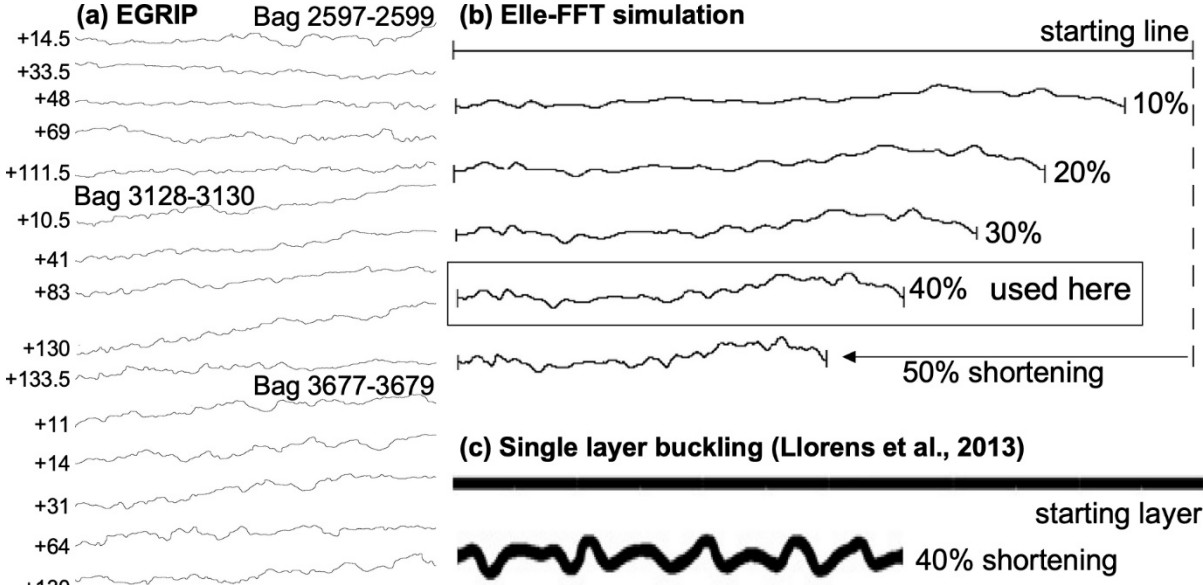

**Figure 6. Fold traces. (a) Traces of folded cloudy-band interfaces in the EGRIP drill core bags 2597, 3128 and 3677. Numbers on the left indicate the distance in cm from the top of the 165 cm-long line-scan image. (b) Traces of an originally horizontal line at different amounts of horizontal shortening in a simulation with Elle-FFT with pure ice Ih (n=4) and initially a horizontal alignment of basal planes. (c) Example from fig 7c in Llorens et al. (2013) of single-layer buckling in a power-law material (n=3).**

In case of folds in an anisotropic material, simulated with Elle-VPFFT, a straight horizontal line, consisting of 102400 nodes, at a chosen level in the model, was subjected to deformation according to the velocity field that Elle-VPFFT records for each deformation step, up to the strain for which the power spectrum was to be calculated (Fig. 6b). The resulting line was then





divided in 256 *x,y*-coordinates that are equidistant in the *x*-direction by interpolating between the original nodes, after which the procedure is the same as for digitised folds in the cloudy bands.

Linear least-squares best-fits were applied to the log(*A*) versus log(*l*) data of the power spectra to obtain the scaling exponent *s* (Eq. 2), using the freeware Past4, except for the single-layer buckle fold where Eq. (2) does not apply.


**(a) Isotropic**: numerical simulation of folded layer Llorens et al. (2013)

**(b) Anistropic**: natural example of folded biotite schist

$A=0.029\cdot\lambda^{1.06}$

**(c) Anistropic**. numerical Elle-VPFFT simulation

$A=0.013\cdot\lambda^{1.01}$

**Figure 7. Power spectra of analysed folds. The vertical axis showing the amplitude is linear in the left column and logarithmic in the right column. (a) Numerical simulation of a single layer in a homogeneous and isotropic softer matrix**



**(Llorens et al. 2013). The power spectrum shows a distinct peak at a wavelength of 8 length units with a total length of**

**the fold train of 60 units. The layer had an initial length of 100 units. (b) Folded foliation in the biotite schist from Puig**

**Culip shows an approximately self-similar power-law power-spectrum. (c) Numerical simulation with Elle-VPFFT of**

**the folding of intrinsically anisotropic, pure ice that has an initial strong alignment of basal planes parallel to the**

**shortening direction. Dots are the average of data sets in the model, while grey lines show the one standard deviation**

**variation. The power spectrum follows an approximately self-similar power law from a wavelength of about five**

**elements-widths of the initial 256x256 model.**

**Table 1. Results of power-law fits to the different fold sets.**

| Fold set | $\lambda$ range | Best fit $s$ | $s$ range | $r^2$ |
|---|---|---|---|---|
| Biotite schist | 0.25 - 47, relative to original length of 100 | 1.057 | 0.976 - 1.152 | 0.90 |
| Elle+VPFFT | 5 - 153 elements | 1.010 | 0.935 - 1.087 | 0.68 |
| Cloudy bands | 1 - 65 mm | 0.965 | 0.884 - 1.047 | 0.58 |
| NEGIS | 100 m - 10 km | 1.487 | 1.39 - 1.62 | 0.86 |
| NEGIS + cloudy bands | 3 mm - 10 km | 0.827 | 0.80 - 0.86 | 0.99 |

## 4. Results

### 4.1 Single-layer buckle folds simulation

Llorens et al. (2013) used finite-element modelling for folding of a competent single layer in a homogeneous softer matrix. They used an isotropic power law rheology relating strain rate ($\dot{\varepsilon}$) to differential stress (s):

$$\dot{\varepsilon} = B \cdot \sigma^n \tag{3}$$

Here $B$ is the pre-exponential factor and $n$ the stress exponent that was set at $n$=3. We show (Fig. 7a) the power spectrum for

40% layer-parallel shortening of a layer of original length 100 and unit thickness. The layer was made 25 times stronger than the matrix by setting $B_{matrix} = 25 \cdot B_{layer}$. The resulting fold (Fig. 6c) shows 8-9 distinct antiforms in the fold train, but no clear folds with other wavelengths. The power spectrum (Fig. 7a) shows a distinct peak at 7.5 times the original layer thickness for a fold-train length of 60 after 40% shortening. The initial wavelength was thus about 11-12 times the layer thickness.

### 4.2 Folded biotite schist

The folded foliation in the biotite schist from Puig Culip (Fig 3b) shows a power spectrum with a steady increase of the amplitude with the wavelength (Fig. 7b). A power-law best fit results in an exponent of approximately $s$=1, implying that the amplitude is linearly proportional with the wavelength (Eq. 2) and the folds are approximately self-similar.



### 4.3 Elle-FFT simulation

We analysed ten equally spaced originally horizontal lines in the model of folding in ice with aligned basal planes (Fig. 8). We

chose a finite strain of 40% shortening, as the lines folded to achieve relative arc lengths of 1.14±0.2, comparable to those

obtained from the cloudy bands (see below). Initially horizontal lines are folded with various wavelengths. Medium to large

folds can be traced along their axial planes over many lines, suggesting that the folds are more harmonic than those in the

cloudy bands if the model is assumed to have a comparative width as the EGRIP drill core.


The power spectrum shows a steady increase in power with wavelength from $l \approx 5$ initial element widths (Fig. 7c), which is in

the order of the mean grain width after 40% horizontal shortening and $A_{(l)}$ increases approximately linear with $l$, i.e., s ≈ 1.

This means that the folds are self-similar: their shape is independent of their wavelength.

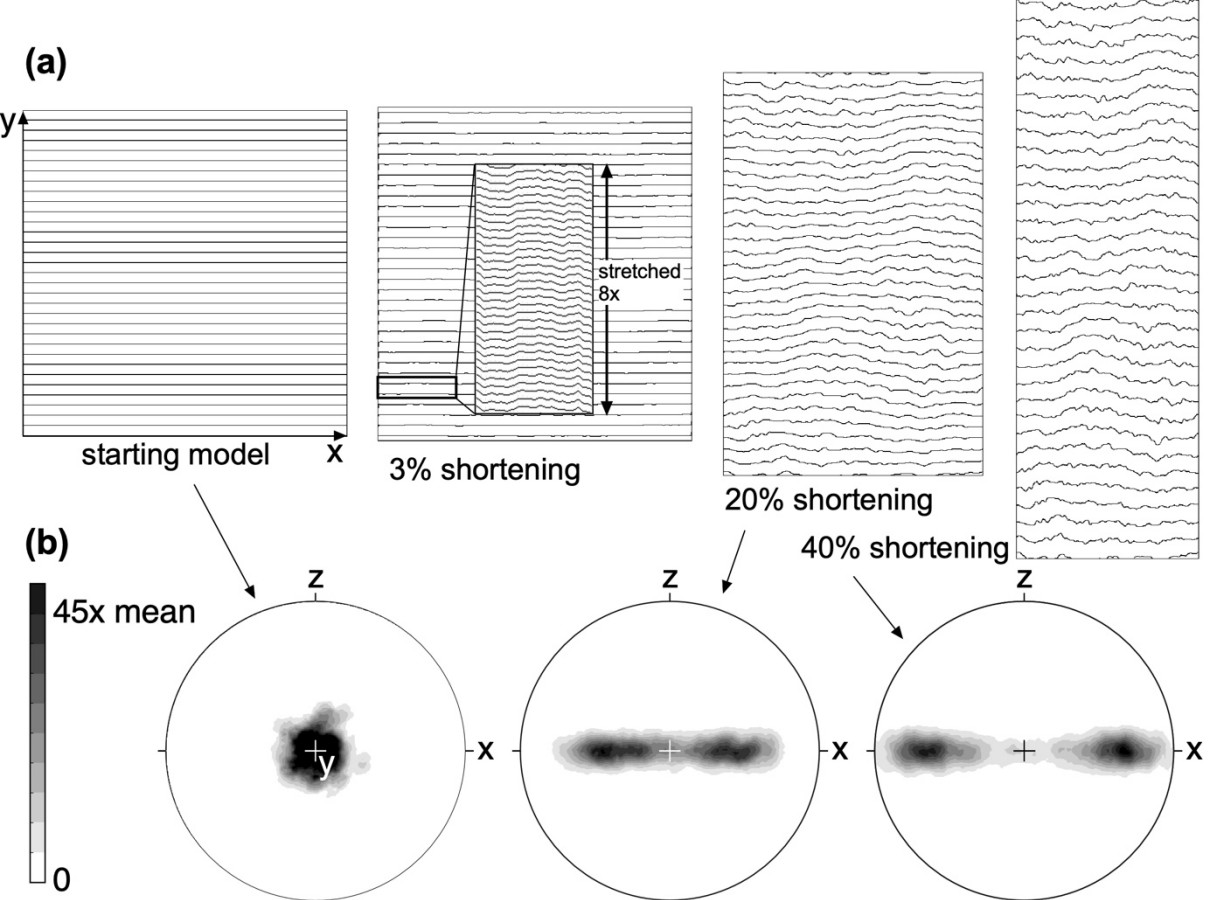


**Figure 8. Result of the Elle-VPFFT simulation showing the model at 0, 3, 20, and 40% horizontal (x-axis) shortening that is compensated by vertical (y-axis) stretching in plane strain. (a) Passive markers originally aligned to the**





horizontally aligned basal planes. At 3% shortening, folds can barely be discerned. At 8x vertical stretch initial folds become visible. These have relatively straight limbs and sharp hinges compared to folds at higher strains. (b)

Stereographs of the c-axis orientations looking down along the vertical y-axis. Plots were created with Stereonet by F.W. Vollmer, using orientations of 2500 randomly selected elements out of a total of 65536 elements.

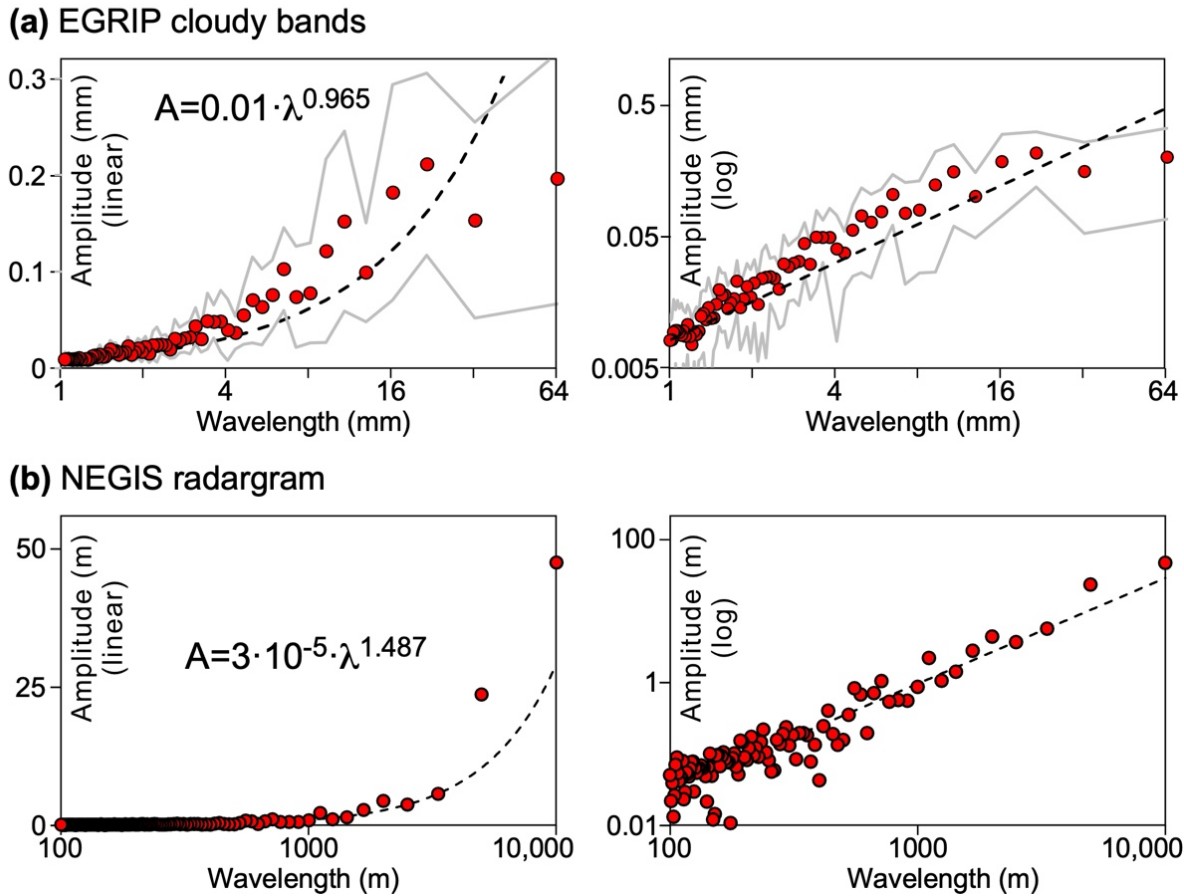

**Figure 9. Power spectra for folds in ice. (a) Average power spectrum of all 15 cloudy band interfaces (red dots), shown together with plus/minus one standard deviation (grey lines) of the variation in powers. The data roughly follow a self-**

**similar power law up to $l \approx 20$ mm. (b) Power spectrum for the internal reflection horizon (IRH) in the radargram that intersects the EGRIP drill core at ca. 1720 m depth (Figure 5) traced over a distance of 10 km perpendicular to NEGIS' flow direction. A power-law best fit for $l \geq 100$ m results in a scaling exponent of $s \approx 1.5$. The vertical amplitude axis is linear in the left column and logarithmic in the right column.**





### 4.4 Cloudy bands

Cloudy bands have a variety of thicknesses, ranging from about a mm to a few cm (Fig. 1 and 4). However, it is difficult to define the thickness of one band, as what appears like one dark or bright band may itself be composed of several thinner bands of different brightness (Stoll et al., 2023). All interfaces are folded on the mm to cm scale (Figs. 1 and 4). Folding is most

conspicuous in the interfaces of very bright and very dark bands. The folds are upright with mostly vertical axial planes (Fig. 1b), although some zones with tilted axial planes were observed (Westhoff et al., 2021). Folds are disharmonic, meaning that individual axial planes can rarely be traced from one interface to another, i.e., for more than about 5 mm (Fig. 1). This means that the folds in individual interfaces appear independent of those in the next. Relative arc length ratios of the 15 folded interfaces are on average 1.15 (± 0.03 standard deviation).


The individual and average power or amplitude spectra of the 15 folded interfaces at the three selected depths show no significant differences. We therefore averaged the powers for each wavelength, as shown (Fig. 9a). We see that the amplitudes first increase up to a wavelength of about 2 mm, followed by a shallower, linear ($s \approx 1$) increase up to about $l = 20$ mm. The amplitudes of the largest two wavelengths are below the trend, but it cannot be ascertained whether this is significant, or merely

due to the large variation in amplitudes. No dominant wavelength well below 65 mm was observed.

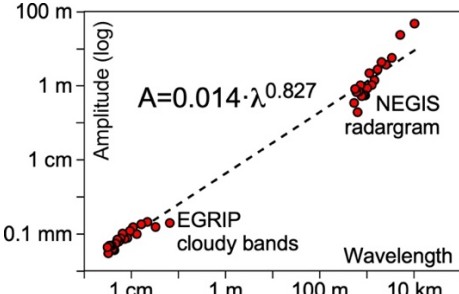

**Figure 10. Combined power spectra of cloudy band interfaces and the internal reflection horizon (IRH), showing the 40 largest wavelengths of each spectrum. A power-law best fit through these data gives an exponent $s = 0.827$, suggesting**

**that folding is self-affine (Fig. 2d), with large folds relatively flat compared to small folds.**

### 4.5 Large-scale folds inside NEGIS

The 1720 m depth layer in NEGIS shows a power-law amplitude-wavelength trend with $s \approx 1.5$ upwards from $l = 100$ m (Fig. 9b). This indicates that there is no characteristic wavelength and that small folds are flatter in shape than large folds. However,

the fold trace may be too smooth on the small scale to correct for small vertical shifts and steps in reflector depths that are artifacts related to surface elevation variations. Underestimating amplitudes at low wavelengths would artificially increase the exponent $s$.




## 5 Discussion

Folds in the cloudy bands do not show a dominant wavelength in the wavelength range shorter than the width of the drill core section (Fig. 9a). The power spectra of the 15 sampled fold traces all overlap and show no significant variation between them, although the cloudy bands immediately adjacent vary in thickness from 1 to >10 mm. Both observations suggest that the folding has no characteristic length scale. In case of Biot-type buckle folding, this would be the thickness of the layers, here the cloudy bands. For Biot-type buckle folding one would expect shorter wavelength of the folds in the ca. 1 mm thick cloudy

band 3677 + 11 than in the ten times thicker one at 3677 + 14, which are only 3 cm apart. Multiple-wavelength folds can develop in multi-layers (e.g., Fig. 11 in Frehner and Schmalholz, 2006). These form by the addition of different characteristic wavelengths of layers with different thickness and/or rheology. However, harmonic folds are then expected with axial planes extending across several layers that contributed to the multi-wavelength folds. In case of the analysed cloudy bands this is not the case, as the axial planes rarely extend for more than 5-10 mm. These observations suggest that the folds in the cloudy bands

are not the result of Biot-type buckling due to rheological differences between individual cloudy bands, where a high viscosity contrast between layers (≥ 25) is required for folding (Llorens et al., 2013).

The cloudy band folds resemble folds in the biotite schist (Figs. 3b, 7b) and the numerical ELLE-FFT folds in ice with a strong CPO (Figs. 7c, 8) much more than Biot-type buckle folds (Fig. 3a, 6c, 7a). This holds for both a visual assessment and for the

power spectra. Shortening parallel to an intrinsic initial (before onset of folding) mechanical anisotropy due to a CPO is therefore the preferred mechanism to explain the observed folding in the cloudy bands, as was already suggested by Jansen et al. (2016). We observe a similar lack of a characteristic wavelength in the large-scale folds inside NEGIS (Fig. 9). This supports the suggestion by Bons et al. (2016) and the numerical simulations by Zhang et al. (2024) that such large-scale folding is also due to shortening parallel to the CPO-induced anisotropy.


The cloudy bands and internal reflection horizons are thus passive material planes whose folds reveal the heterogeneity of deformation due to CPO-induced anisotropy. This CPO was probably a vertical single maximum of the c-axis orientations. However, folding of course changes this CPO, resulting in a girdle with (Fig. 8b) or without (Stoll et al. 2023) two maxima. The passive marker planes do not reflect the current CPO, but the cumulative heterogeneous strain resulting from the evolving

CPO.

Folding of an intrinsic anisotropy has no typical length scale, which explains the close to self-similar ($s \approx 1$) power spectra of the cloudy band interfaces. If the folds were perfectly self-similar and the self-similar range would extend to the 10 km scale that we observe in the traced IRH in the radargram, we would expect folds about seven times taller than the folds we observe

(Fig. 5).



A power-law best fit through the combined cloudy-band and radargram folds results in a scaling exponent of $s \approx 0.8$ (Fig. 10). This would mean that the folds gradually get flatter with increasing scale. Unfortunately, we do not have sufficiently detailed observations in the length-scale range from 10 cm to about 100 m. It is therefore possible that the power spectra between the
small and large scale show a break. Another possibility is that the scales are related and that the folds are self-affine (Fig 2d). Reasons for this could be the effect of the ice-sheet surface, where gravity and ice precipitation counteract the development of surface topography due to folding (Waddington et al., 2001; Zhang et al., 2024). An additional effect could be the bedrock topography or bedrock processes (Bell et al., 2014; Wolovick et al., 2014; Leysinger Vielli et al., 2018) that impose additional folding or fold-amplitude modifications of the anisotropy-induced folds. Unfortunately, the radargrams are of poor quality for
the detailed fold analyses that are applied here. A more systematic analysis of the large-scale folds is needed but is outside the scope of this study.

Having argued that folding in cloudy bands in ice sheets is the result of shortening of an intrinsic anisotropy, we now briefly address the degree of anisotropy. Materials with a very strong transverse isotropy tend to form kink or chevron type folds
(Cobbold et al., 1971; Ramsay, 1974; Nabavi and Fossen, 2021). Such chevron folds, or "fabric stripes", have been described in ice with a strong CPO (Alley et al., 1997; Jansen et al., 2016). With decreasing degree of anisotropy folds become more rounded and axial planes are oriented perpendicular to the direction of maximum shortening (Cobbold et al., 1971), as is the case in the folds in cloudy bands shown here (Fig. 4). The ELLE-VPFFT simulation started with a strong single-maximum CPO (Fig. 8b). Initial (3% shortening) folds are relatively chevron-like. Such a low strain is comparable with the model
proposed by Jansen et al. (2016) where chevron-type fabric stripes that develop in simple shear with a c-axes single maximum almost perpendicular to the shear plane. The situation is different in pure shear shortening that leads to much higher of cloudy bands initially perpendicular to the preferred c-axis orientation. Here the CPO evolves (Fig. 8b) together with the folding of the cloudy bands, leading to more rounded folds at all scales.

In the absence of any other factors that control folding, the proportionality factor ($l_0$) in Eq. (2) should only depend on the type of anisotropy, its intensity, and the amount of shortening. This implies that it should be possible to determine the actual anisotropy of ice with folds (e.g. the EGRIP drill core) if the initial CPO can be constrained and the amount of shortening parallel to the folded planes (e.g. the cloudy bands) is known. This could be a helpful independent estimate of the anisotropy of flowing ice, and, hence, its effective hardening or softening due to the flow field, in addition to other studies (Gerber et al.,
2023). This is, however, well beyond the scope of this study.





## 6 Conclusions

We used power spectra of fold traces to determine the mechanism for folding in ice sheets. Biot-type buckle folds due to rheological contrasts between layers have a characteristic length scale, related to the layer thickness and the rheological contrast

between the layers that are internally isotropic. Numerical simulation of ice with a strong alignment of basal planes parallel to the shortening direction resulted in the development of self-similar folds with a power-law power spectrum. This is to be expected as anisotropy has no length scale. Self-similar folds were also observed in folded biotite schist and in cloudy bands in the EGRIP drill core. We therefore conclude that small-scale folds in cloudy bands are due to shortening parallel to a strong anisotropy as a result of the lattice preferred orientation with initially horizontally aligned basal planes. Combining the small

cloudy band folds and large NEGIS-scale folds resulted in a self-affine trend, where largest folds are relatively flat. This may be caused by additional boundary conditions, such as vertical flattening and bedrock irregularities, that modify the anisotropy-induced folds on the large scale.

## Code availability

The updated version of Elle+FFT software package from Hao et al., (2023) can be downloaded from https://doi.org/10.5281/zenodo.10259841. We recommend using the Elle+FFT software package by Singularity container under Linux (e.g. for Ubuntu 20.04). Additional information on the software can be found in the Appendix section of PhD-thesis of Dr. Florian Steinbach, which can be downloaded from the library of Tübingen University (https://publikationen.uni-tuebingen.de/xmlui/handle/10900/76435). The Elle+VPFFT input files (ASCII text files) to rerun the simulations used in this

paper and can be downloaded as supplementary data.

## Data availability

Visual stratigraphy data from the EastGRIP ice core are available at https://doi.org/10.1594/PANGAEA.925014 (Weikusat et al., 2020). The radio-echo sounding data shown in Figure 5 (profile IDs: 20180508_06_004 and 20180514_03_001 from

AWI's EGRIP-NOR-2018 survey are available under https://doi.org/10.1594/PANGAEA.928569 (Franke et al., 2021, 2022b). Input files for the Elle+VPFFT numerical simulation shown here (folder supplement_Elle_VPFFT_files) are provided as Supplementary Data to this article. The data contains x-y coordinates of all analysed fold trains as tab-delimited ASCII text files.



**Author contributions**

PDB conceived the study, performed most analyses, and wrote the manuscript. JW, NS and IW acquired and processed visual stratigraphy data. GML, YH, and YZ carried out numerical simulations and contributed to modelling efforts. SF processed and contributed radar data. All authors jointly contributed to the discussion of results and reviewed the manuscript.

**Competing interests**

The authors declare that they have no conflict of interest.

**Acknowledgements**

We acknowledge the use of software from Open Polar Radar generated with support from the University of Kansas, NASA grants 80NSSC20K1242 and 80NSSC21K0753, and NSF grants OPP-2027615, OPP-2019719, OPP-1739003, IIS-1838230, RISE-2126503, RISE-2127606, and RISE-2126468. EastGRIP is directed and organised by the Centre for Ice and Climate at the Niels Bohr Institute, University of Copenhagen. It is supported by funding agencies and institutions in Denmark (A. P.
Møller Foundation, University of Copenhagen), USA (US National Science Foundation, Office of Polar Programs), Germany (Alfred Wegener Institute, Helmholtz Centre 475 for Polar and Marine Research), Japan (National Institute of Polar Research and Arctic Challenge for Sustainability), Norway (University of Bergen and Trond Mohn Foundation), Switzerland (Swiss National Science Foundation), France (French Polar Institute Paul-Emile Victor, Institute for Geosciences and Environmental research), Canada (University of Manitoba) and China (Chinese Academy of Sciences and Beijing Normal University).


Julien Westhoff acknowledges funding from Villum Investigator project IceFlow (grant no. 16572). Yu Zhang was funded by the China Scholarship Council (Grant number202006010063). Yuanbang Hu was funded by the China Scholarship Council (Grant number 202008510177). Steven Franke was funded by the Walter Benjamin Programme of the Deutsche Forschungsgemeinschaft (DFG, German Research Foundation; project number 506043073). Nicolas Stoll acknowledges
funding from the Programma di Ricerche in Artico (PRA).

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
