# Peer review of "Folding due to anisotropy in ice, from drill-core-scale cloudy bands to km-scale internal reflection horizons"

_EGUsphere, 2024_

## Referee Comment (RC1)

Review of:

**Folding due to anisotropy in ice, from drill core-scale cloudy bands to km-scale internal reflection horizons**

Submitted to: *The Cryosphere*

**Summary.**

Bons et al. use measurements of layer folding in ice, across a wide range of spatial scales, to test hypotheses for the mechanism of layer folding. Based on the frequency spectra of their observed folds, they find that folds which develop in the cross-flow direction of an ice stream (here, in Northeast Greenland) are the result of intrinsic anisotropy in ice. This is distinct from the conventional "Biot-type" fold which results from rheological contrasts (e.g., in a metamorphic rock with an intrusion). I find this study an important contribution to a developing body of work on this topic. The article is extraordinarily well written, fun to read, and a great fit for *The Cryosphere* after these minor revisions.

**General Comments.**

My most significant comment is that you could do more to draw a distinction between the two mechanisms of folding:

- Is there not a good way to include the frequency spectra analysis for biot-type folds? You have the model in Figure 7a, which is great, but for the sake of parallelism with the anisotropic fold (where you use the schist) can you also include some analysis of the image in Figure 2a? or use a different image of the Biot-type folding?
- Figures 2 (c and d), 3 (a and b), 6 (b and c), and 7 (a-c) are all drawing a contrast between the two fold mechanisms. Some parallelism between the figures (which is on top/bottom or left/right) as well as some annotations/labels for biot-type and anisotropic to make it abundantly clear the distinction you are trying to draw. That would all help me as a reader.

Is the horizontal resolution of the radar really 15 meters? Franke et al. (2022) say that the PRF is 10 kHz and aircraft velocity is 260 km per hour (72 m/s), so >100 pulses per meter. There is probably some onboard stacking, so less recorded traces, but even so, at a lowerlevel data product there must be better along-track resolution than 15 meters. The reason I think this is important is that you may be able to fill in the gap in figure 10 (extending the radar layer to shorter wavelengths) which would make the entire manuscript stronger in my opinion.

Otherwise, you could consider other radar systems for this or future work? I know there have been recent ground-based surveys at EGRIP with the CReSIS accumulation radar (more like cm range resolution and probably ~cm along-track resolution as well).

The fold amplitude of the radar layer changes significantly with layer depth. Since these layers have harmonic folds, I think that this amplitude change would only shift your power spectra up/down uniformly, not change the exponent you derive. However, it would change the relative placement in Figure 10. Do you think that the nature of the cloudy band folds would change significantly with depth as well? Perhaps expand on this point with a could sentences in the discussion?

In the caption of Figure 8 you mention down-glacier (y-axis) extension in the context of the fabric development. I believe that strain component would not be included in your 2-d model. I don't think that it needs to be, but adding a couple sentences somewhere on how you think this may or may not affect your results would be useful.

**Specific Comments.**

L33 – space after comma

Eq1 – are you intentionally switching between l and lambda for wavelength?

Eq2 – same as in eq1, suggest using lambda in the text

L138 – I would reframe the start of this paragraph to focus on ice instead of schists. "Ice 1h is comparable to micas" rather than the reverse, since ice is the material of focus in this article.

L147 – citation to 4? Or referencing figure 4?

L148 – I am confused about the reference to figure 2 here.

L148-149 – I would argue that the line scan numbers are not very useful unless there is a history of using them as a convention that I am not familiar with.

L160 – As with the line scan numbering, I argue that citing (Franke 2022) is enough, the survey name doesn't mean much.

L183-184 – Is there a citation for the 16x increase in B?

L189 – is there a reference for the Potts model?

3.2.2 "measured" fold analysis?  To draw a distinction from the previous section, it caught me off guard for some reason.

L201-202 – Is there a real reason to use the bag numbers instead of depth?

L219 – Fig 5 comes after 6 in the text

L219 – I assume that you are not implying the radar layer is representative of one of the layers in the ice core, could be worth stating that explicitly.

L238-239 – is this meant to be its own paragraph?

L262 – 7.5x or it says 8x in the figure caption, choose one for consistency.

L308 – Are you interpreting the contrast at 2 mm to be significant? Or within the uncertainty of the measurement. If you think it is representing some physical process, I would expand on it and make it more clear in the figure. Otherwise, I think you can ignore it.

L320-322 – I believe what you are saying here is that you are limited by the range resolution of the instrument? If so, you could consider adding more data for this or future studies (I expanded on this in general comments above)

L364-365 – The data are not really poor quality (the opposite in fact). I would say the range resolution is not sufficient for your objective.

**Figures.**

Figure 6 – I would keep the data and model results separate, 6a belongs in figure 4 in my opinion.

7&9 – the linear plots don't add much in my opinion. Power spectra are almost always plotted in log space anyway

Figure 7 – As I noted in general comments above, I think you could do more to emphasize the differences you want a reader to see here. Add annotations that point out the peak power at $l=8$ in (a). Perhaps consider grouping (b and c) separate from (a) to make it abundantly obvious that the reader is meant to be contrasting those.

Figure 8 –Are there any important differences between this figure and 6b aside from the inclusion of the stereographs in (b)? If the stereographs are important, this figure could emphasize those, otherwise, I think anything being demonstrated in (a) can be summarized in 6b.

Figure 9a – Are the two points below your regression line because they are approaching the length of the image? In line 208 you say you 65 mm is the length of the image, so I am not surprised that the power drops off as you approach that wavelength.

---

## Author Response (AR1)

This document contains the reviewers' comments, shown in black font. replies to these comments are shown in blue font. New or modified sections of text in the manuscript are shown in red font.

The sentence "Both photographs by Paul Bons." was added to the caption of Fig. 3 to clarify authorship of the images.

No changes needed to the originally submitted supplement.

**REVIEWER 1**

**Summary.**

Bons et al. use measurements of layer folding in ice, across a wide range of spatial scales, to test hypotheses for the mechanism of layer folding. Based on the frequency spectra of their observed folds, they find that folds which develop in the cross->low direction of an ice stream (here, in Northeast Greenland) are the result of intrinsic anisotropy in ice. This is distinct from the conventional "Biot-type" fold which results from rheological contrasts (e.g., in a metamorphic rock with an intrusion). I find this study an important contribution to a developing body of work on this topic. The article is extraordinarily well written, fun to read, and a great fit for The Cryosphere after these minor revisions.

**General Comments.**

My most significant comment is that you could do more to draw a distinction between the two mechanisms of folding:

• Is there not a good way to include the frequency spectra analysis for biot-type folds? You have the model in Figure 7a, which is great, but for the sake of parallelism with the anisotropic fold (where you use the schist) can you also include some analysis of the image in Figure 2a? or use a different image of the Biot-type folding?

The problem with natural folds in ductile rocks is that ductile rocks usually have a crystallographic orientation and/or cleavages. These may influence folding on top of the (non-linear) viscosity contrast between a layer and its matrix. We therefore chose the numerical fold of Llorens et al. (2013) because this is ideal Biot-type folding. However, to show the effect of ideal single-wavelength folding, we applied the fold analysis to a cosine wave (as in Fig. 2a) and can add it as fig. 7a (with b-c shifting to c-d):

(a) Cosine wave with wavelength 140 pixels

For this we needed to modify the text. We revised Section 3.2.2 from line 200: We created bitmap images of the folds to analyse, with the fold trace drawn as a black line of a few pixels' width, roughly parallel to the x-axis. A script selected the y-coordinates of the line for each x-coordinate along the trace. The equidistant x,y-data were then detrended by subtracting a linear least-squares best fit through the x,y-data. The detrended series of y-data was then subjected to a discrete Fourier transform using the routine four1() of Press et al. (1992). The power spectrum was obtained by taking the square root of the sum of the squares of the real and imaginary parts

of the transform for each wavelength. An example of the result is given in Fig. 7a, in which the input was a graph of a cosine function with a wavelength of 140 pixels and an amplitude (as defined in Fig. 2a) of 50 pixels.

The FFT-routine gives the power for wavelengths  $\lambda$ =W/n, where W is the width of the y-series and n is a whole number greater or equal to one. The nearest wavelength to  $\lambda$ =140 pixel of the cosine signal is 142.9 pixel (n=7, for a series of 1000 pixels length). The amplitude obtained for this wavelength is 35 instead of 50 pixels (Fig. 4a). This is for two reasons: (i) the wavelength is not exactly that of the real wavelength of the cosine signal, and (ii) rounding errors in the digitisation of the signal affect the whole signal. Whereas wavelength other than 140 should be zero, we see they are not in the FFT-analysis, but still distinctly lower than the dominant wavelength.

In the process, we discovered that Fig. 3a was not referred to in the text. After Eq. (1), line 116 we added:

For a certain  $\eta / \eta_m$ -ratio, the dominant wavelength is proportional to the layer thickness, which explain why the folds have varying wavelengths in the folded aplite dyke shown in Fig. 3a.

• Figures 2 (c and d), 3 (a and b), 6 (b and c), and 7 (a-c) are all drawing a contrast between the two fold mechanisms. Some parallelism between the figures (which is on top/bottom or left/right) as well as some annotations/labels for biot-type and anisotropic to make it abundantly clear the distinction you are trying to draw. That would all help me as a reader.

We added "Biot", "Self-similar", or "Self-affine" prominently to figures 3, 7, 9.

• Is the horizontal resolution of the radar really 15 meters? Franke et al. (2022) say that the PRF is 10 kHz and aircraft velocity is 260 km per hour (72 m/s), so >100 pulses per meter. There is probably some onboard stacking, so less recorded traces, but even so, at a lower-level data product there must be better along-track resolution than 15 meters. The reason I think this is important is that you may be able to fill in the gap in figure 10 (extending the radar layer to shorter wavelengths) which would make the entire manuscript stronger in my opinion.

Broadly speaking the horizontal resolution of the final product used here is 15 meters due to a trade-off between signal-to-noise ratio (SNR) and the limitations of the system. It is correct that the radar system transmits with 10 kHz, however incoherent onboard stacking already occurs (amount depends on the bandwidth) to reduce the data rate and increase (SNR). This leaves  $\sim 500$  Hz raw data for processing. During processing the process with the highest computational demand is the fk-migration and prior to this step along-track stacking is performed to 2.5 m to reduce the computational costs and to increase SNR (resulting in the 15 m along-track resolution of the final product). It is possible to reduce along-track stacking prior to fk-migration to 1m for example, which will end up in a final along-track resolution of  $\sim 6$  m. However, a higher number of traces and, thus, a larger SAR aperture, would not substantially improve the visibility of specular reflections (IRHs) and might even reduce the clarity of the signal due to reduced SNR due to more noise. Hence, we face a tradeoff between a smaller along-track spacing and less coherence of IRH signals.

Another limiting factor is the vertical resolution given by the system bandwidth (30 MHz), which is here ~4.3 m. Even if it would be possible to increase along-track resolution, we see little total improvement to resolve finer IRH folds due to the limitation in vertical resolution.

As the reviewer suggested, one option for future studies would be to use a system with a broader bandwidth to increase vertical resolution and a ground-based system with a higher number of recorded traces.

Otherwise, you could consider other radar systems for this or future work? I know there have been recent ground-based surveys at EGRIP with the CReSIS accumulation radar (more like cm range resolution and probably ~cm along-track resolution as well).

We agree with the reviewer that, in principle, a higher-resolution radar product would enable detecting smaller-scaled folds. However, we'd disagree with the reviewer that neither the CReSIS accumulation radar nor recent ground-based radar surveys at EGRIP have a cmresolution. Following Gerber et al., 2025 (https://doi.org/10.5194/tc-19-1955-2025) the along-track resolution of their ground-based 300 MHz bandwidth radar is "after processing is approximately 25–30 m". Their bandwidth of 300 MHz would allow a range resolution of ~ 30-50 cm, depending on the processing. For the CReSIS accumulation radar systems, the trace spacing is in a similar range as of our radar data product (if SAR processing is applied). Moreover, the downside of the accumulation radar is that only the first ~ 300 m below the ice-sheet surface can be sounded. The depth of our IRHs is deeper as folding in the stratigraphy is stronger at greater depth (e.g., 1720 m).

In conclusion, we follow the reviewer that better-resolution products are potentially available, but do not represent an improvement of several magnitudes in resolution (cm along-track and range resolution). We added the following paragraph to the discussion addressing this comment and the comment above:

"Looking forward, future studies could benefit from higher-resolved radar data products to better characterize smaller-scale folds in internal reflection horizons. The data set used in this study was limited to a vertical resolution of ~4.3 m due to the system bandwidth of 30 MHz. Future deployments using broader bandwidth systems and optimized acquisition and processing strategies to increase range and along-track resolution could extend the detectable wavelength range of folded horizons. This would help bridge the resolution gap between drill-core observations and radar-imaged folds, thereby refining our understanding of anisotropy-induced deformation across scales."

The fold amplitude of the radar layer changes significantly with layer depth. Since these layers have harmonic folds, I think that this amplitude change would only shift your power spectra up/down uniformly, not change the exponent you derive. However, it would change the relative placement in Figure 10. Do you think that the nature of the cloudy band folds would change significantly with depth as well? Perhaps expand on this point with a could sentences in the discussion?

It is correct that the large-scale power-spectra mostly shift up and down, depending on depth. We suggest showing the spectrum of a shallow IHR at 475 m and that of the bedrock: **(b)** NEGIS radargram

**(b) NEGIS ladalylalli**

The similarity in large-scale fold shape and power spectra suggests a strong bedrock control, with the bedrock shape transferring upwards with decreasing amplitudes. We would modify figure 10 to highlight this:

We reply to the nature of the cloudy bands below.

In the caption of Figure 8 you mention down-glacier (y-axis) extension in the context of the fabric development. I believe that strain component would not be included in your 2-d model. I don't think that it needs to be, but adding a couple sentences somewhere on how you think this may or may not affect your results would be useful.

Elle+VPFFT is strictly speaking 2-dimensional, which excludes including the layer parallel extension. The point of the Elle+VPFFT modelling is not to simulate the cloudy bands, but to show that shortening parallel to a (strong) anisotropy leads to self-similar folds. As mentioned in Jansen et al. (2024), amplification of such folding decreases as the CPO changes with progressive strain. We therefore do not expect much variation in fold spectra of cloudy bands in the column of ice older than about 2000 yrs that experienced the same strain. We did not check this, as deepest core images were not yet available when this work was done.

**Specific Comments.**

L33 – space after comma OK

Eq1 – are you intentionally switching between I and lambda for wavelength? No, our mistake Eq2 – same as in eq1, suggest using lambda in the text Of course

L138 – I would reframe the start of this paragraph to focus on ice instead of schists. "Ice 1h is comparable to micas" rather than the reverse, since ice is the material of focus in this article. Good point; changed.

L147 – citation to 4? Or referencing figure 4? The "4" should not be there at all and is removed.

L148 – I am confused about the reference to figure 2 here. Should be Fig. 4 in Westhoff et al. and is now corrected. Sorry.

L148-149 – I would argue that the line scan numbers are not very useful unless there is a history of using them as a convention that I am not familiar with. These numbers have to do with the bag numbers, and then make sense. We prefer to keep them.

L160 – As with the line scan numbering, I argue that citing (Franke 2022) is enough, the survey name doesn't mean much. We beg to differ, as we think it is correct to mention the actual survey. The survey name might help a reader when they reach out to the Alfred Wegener Institute requesting data.

L183-184 – Is there a citation for the 16x increase in B? It is based on Duval et al. (1983), cited before. It should be mentioned here again. It is based on the ratio of the stress at a certain strain rate for basal and non-basal slip. We added the reference and a short explanation.

L189 – is there a reference for the Potts model? "Potts model" is a general term for this class of cellular automata. The model itself was written for this study. It is not really relevant how it works exactly, as the purpose is to just get a foam texture with equidimensional grains. 3.2.2 "measured" fold analysis? To draw a distinction from the previous section, it caught me off guard for some reason. We are sorry, but we do not understand this comment. In 3.2.2 we deal with the fold analysis as part of the "Methods".

L201-202 – Is there a real reason to use the bag numbers instead of depth? Bags are the actual core sections. To look up the core, one needs the bag number. Depths are derived and may even be corrected at a later stage. Bags should therefore be mentioned.

L219 – Fig 5 comes after 6 in the text. This was checked. No need for changes.

L219 – I assume that you are not implying the radar layer is representative of one of the layers in the ice core, could be worth stating that explicitly. Yes, that is the accepted idea. We can clarified this: The radargram shows a traceable reflector that crosses the EGRIP drill core at about 1720 m depth, which is the depth of bags 3128-30 from which cloudy bands were analysed. This does not mean that the reflector is directly comparable to any single cloudy band that is observe3d in the core at this depth.

L238-239 – is this meant to be its own paragraph? No, there is indeed no need for that. Changed.

L262 – 7.5x or it says 8x in the figure caption, choose one for consistency. Well spotted! We changed both to "ca. 7.5 times"

L308 – Are you interpreting the contrast at 2 mm to be significant? Or within the uncertainty of the measurement. If you think it is representing some physical process, I would expand on it and make it more clear in the figure. Otherwise, I think you can ignore it. Below 2 mm we would regard as insignificant as resulting from the uncertainty of the measurements at these very small and low-amplitude waves. We now leave out the "up to 2 mm" as it may create confusion: We see that the amplitudes show linear ( $s\approx1$ ) increase up to about  $\lambda=20$  mm. L320-322 – I believe what you are saying here is that you are limited by the range resolution of the instrument? If so, you could consider adding more data for this or future studies (I

expanded on this in general comments above). Correct, and we added a paragraph to the discussion mentioning the benefit of better-resolved radar data in the future.

L364-365 – The data are not really poor quality (the opposite in fact). I would say the range

L364-365 – The data are not really poor quality (the opposite in fact). I would say the range resolution is not sufficient for your objective. Fair enough. We can consider modifying the sentence. Moreover, we now addressed the issue of limited range and along-track resolution as well as possible future improvement in that regard in the discussion.

**Figures.**

Figure 6 – I would keep the data and model results separate, 6a belongs in figure 4 in my opinion. We put the fold traces together in fig. 6 to already provide a ready comparison of the different fold traces. This would be lost if the figures are rearranged as suggested. Also note that we partly have a two-tier process, where modelling results are themselves data for the next fold-shape analysis. We prefer to keep fig. 6 as is.

7&9 – the linear plots don't add much in my opinion. Power spectra are almost always plotted in log space anyway. A single-maximum (Biot) is actually best seen in a log-lin plot, which is why we added these too. Although they are, of course, also visible in a log-log plot, we think it is best to keep both (can't harm).

Figure 7 – As I noted in general comments above, I think you could do more to emphasize the differences you want a reader to see here. Add annotations that point out the peak power at I=8 in (a). Perhaps consider grouping (b and c) separate from (a) to make it abundantly obvious that the reader is meant to be contrasting those. We can do this to the extent that we do not overcrowd the figures.

Figure 8 –Are there any important differences between this figure and 6b aside from the inclusion of the stereographs in (b)? If the stereographs are important, this figure could emphasize those, otherwise, I think anything being demonstrated in (a) can be summarized in 6b. This would tighten the paper and we can do this if you and the editor wish so. However, we think it is also good to show the stages of the simulation and that the folding is accompanied by a change of the CPO. We refer to this in the Discussion.

Figure 9a – Are the two points below your regression line because they are approaching the length of the image? In line 208 you say you 65 mm is the length of the image, so I am not surprised that the power drops off as you approach that wavelength. We do not know for sure, but as you mention it is at the limit of the full section length. We therefore refrained from putting much weight on it.

**REVIEWER 2**

Bons et al. investigated the mechanism behind folding observed at various scales within ice sheets, from centimetre-scale cloudy bands in drill cores to kilometre-scale internal reflections seen in radar data. By analyzing the power spectra of these folds, they find that they lack a characteristic wavelength, unlike Biot-type folds resulting from rheological contrasts between layers. The study proposes that this folding primarily arises from the strong mechanical intrinsic anisotropy of ice. This is supported by numerical models of anisotropic ice deformation and the similarity between the power spectra of cloudy bands and folded biotite schist.

The manuscript provides a great overview of fold theory and the authors concludes their finding very clearly, highlighting the potential impact this study can make. I was less content with the description of the methodology and the results. Here the organisation of the text and the figures introduced some confusion. This being said, I think the manuscript has great merit and after some revision (with special care to the mentioned sections) it will be a great contribution to the Cryosphere. Below, I have provided my comments section by section, mirroring the manuscript's organization.

We'd like to thank the reviewer for the interest and generally positive feedback as well as for the time revising our manuscript. Our replies to the comments and the changes we made in the revised text are outlined below.

**Introduction**

L63: Can you specify what are the certain preferred orientations? We added to this sentence: ", typically perpendicular to the finite fattening direction "

Basic fold terminology and theory

Throughout the text "I" is used within the text, while is used in the equations for the dominant wavelength and the proportionality constant. Sorry, Reviewer 1 also noticed this. This is a formatting error and we now use lambda consistently.

Materials and methods

L152-153: This sentence seems like an unnecessary repetition. Correct, this sentence is now deleted.

Instead, after Fig. 4 something is missing that transitions the reader's attention from cloudy bands to large-scale folds observed on radar data. Yes, this is true. We added a sentence to make the link: "Cloudy bands are only visible in drill core with a width of <10 cm. Much larger-scale folds are observed in ice by means of radar data."

3.2.1 Numerical modelling: Since they are also discussed later, I think it would be useful to introduce here the numerical models from Llorens et al. Until L216 it has not been mentioned, but then it becomes a significant part of the results and the discussion. This and the previous comment make it clear that a section is needed at the end of the introduction where we explain the approach taken in this paper. Here we now introduce the different methods and analyses.

L176: mechanical anisotropy. Corrected

L189-190: To clarify this: each cluster contains some number of crystals with the same orientation that is then called a grain. Or a cluster contains some grains with a given orientation distribution? Sorry for the misunderstanding (a comma would already have helped). A grain is defined as a cluster of nodes with initially identical lattice orientation. We rewrote this to make sure this is clear: Using a Potts model, we created 1995 grains that are on average 6x6 unodes in size. All unodes inside one grain initially have the same lattice orientation. The lattice orientation of each grain is randomly drawn from a point maximum (with a standard deviation of 10°) parallel to the vertical extension direction. Easy-glide basal planes are thus preferentially aligned parallel to the horizontal shortening direction, with a standard deviation of 10°.

**Fold analysis**

L 201: It could be useful to say that this was to look at the cloudy bands on ice cores. sentence modified: We used three 2044x31550 8-bit images of cloudy bands in the line scans of the EGRIP drill core with a resolution of 18.6 pixel/mm

I think there is a bit of a mixture of method and results here, with seemingly random order (also inconsistent with the figures) between describing the method and results for ice core, ice radar, numerical anisotropic models and the numerical isotropic models.

Why are Figure 7 and Table 1 in section 3.2.2 and not in section 4?

First of all, we tried to place the figures close to the relevant position in the text, while avoiding large white spaces in the document when the figure would not fit on a page anymore. We assumed that the Cryosphere editorial team would do the final layout. The complication is that the paper is based on field, drill core and radargram images, but also on results of new numerical modelling. This means that we also have to describe the numerical modelling method and results, and then the analysis that applies to these and all images. the logic of the order of text and figures is based on this. As mentioned above, we added a few sentences to better explain the general approach and transition from one section to the next.

Table 1: what do 5-153 elements scale to? The initial model is 256 elements wide. This is shortened to a width of 153. We now clarified this in the table.

**Results**

L210-275: The results of the Elle-FFT models are compared to the cloudy bands, but the cloudy band's results are described later. This is not logical to me. Our intent is to first show Biot theory and self-similar folds due to anisotropy, based on the field observation (biotite schist) and numerical modelling. Having shown the two end members, we then proceed to the cloudy bands. The idea is that the reader will then have the two fold types clearly in mind. This is now explained in new text at the end of the introduction.

Figure 8a – isn't this already shown in Fig. 6b? There is indeed some overlap, but - as mentioned above - fig. 6 puts all different fold traces together for easy visual comparison. Yes, there is some overlap, however, the figures point out different comparisons: fold shape in 6 and fabric in 8. We believe that this provides better clarity for the reader and therefore we decided not to change the figure order or layout in this regard.

Figure 9 - Is it even reasonable to look at the power spectra of wavelengths that are the same as the sample size? No, not really, this was already addressed above

L308: I can't observe the change in amplitude increase before and after the 2 mm wavelength. This was already addressed above

Figure 10: Move this into the discussion. Why are the axes labelled inside the plot? Labels inside the plot is simply to reduce space. New version (see above) has labels outside. *Discussion*

L328: ..folding this would be... - replace this with the characteristic length scale. It's not obvious what "this" is referring to. Good suggestion - done.

L330: which are only 3 cm apart -would it be worth adding that this is not observed? This reference to 3 cm is actually not really relevant. We deleted it.

L349: What do you mean here? Sorry that this was too brief. Originally, the basal planes in the model were, on average, parallel to the horizontal material lines. As the CPO and folds develop, this is no longer the case. The distribution of layer dips does not any more signify the variation of basal plane dips. We rewrote this section (see above)

L351: intrinsically anisotropic layer? No, the material is homogeneous, so not layered, but it is anisotropic.

L377: Not just the CPO evolves with the folding, but also the CPO-induced (macroscopic) anisotropic viscosity. That is true! Changed to: The situation is different in pure shear shortening that leads to stronger shortening of the cloudy bands that are initially perpendicular to the preferred c-axis orientation. In this case both the CPO and the CPO-

| induced mechanical anisotropy evolve (Fig. 8b) together with the folding of the cloudy bands, leading to more rounded folds at all scales. |  |
|--------------------------------------------------------------------------------------------------------------------------------------------|--|
|                                                                                                                                            |  |
|                                                                                                                                            |  |
|                                                                                                                                            |  |
|                                                                                                                                            |  |
|                                                                                                                                            |  |
|                                                                                                                                            |  |
|                                                                                                                                            |  |
|                                                                                                                                            |  |
|                                                                                                                                            |  |
|                                                                                                                                            |  |
|                                                                                                                                            |  |
|                                                                                                                                            |  |
|                                                                                                                                            |  |
|                                                                                                                                            |  |
|                                                                                                                                            |  |
|                                                                                                                                            |  |

---

## Author Response (AR2)

Ejin Banner, August 23, 2025

Dear Editor, dear Carlos,

we are very pleased that the review process is now completed.

Reviewer #1 still found some typos for which we apologise. We corrected all.

The reviewer questioned the hyphening in the title: "Should the title be "drill-corescale"? with a three-word hyphenation? or you could drop the word "drill". I don't have a strong preference, but worth consideration." Here 'drill-core scale' is used as a compound adjective and to our knowledge of English grammar, the individual words should then by hyphenated. We suggest keeping it as is, but we are open to alternative suggestions by the journal.

We added a few words to the short summary and updated the Supplement according to the figure numbers in the current revised manuscript.

There are no copyright issues with any of the figures.

We thank you for handling our manuscript.

Kind regards,

Paul Bons, on behalf of all authors.